# Experimental Study on Capillary Microflows in High Porosity Open-Cell Metal Foams

**DOI:** 10.3390/mi13122052

**Published:** 2022-11-23

**Authors:** Huizhu Yang, Yue Yang, Binjian Ma, Yonggang Zhu

**Affiliations:** School of Mechanical Engineering and Automation, Harbin Institute of Technology (Shenzhen), Shenzhen 518055, China

**Keywords:** wicking, capillary performance, metal foams, superhydrophilicity, heat pipes

## Abstract

Metal foams have been widely used in heat pipes as wicking materials. The main issue with metal foams is the surface property capillary limit. In this paper, a chemical blackening process for creating a superhydrophilic surface on copper foams is studied with seven different NaOH and NaClO_2_ solution concentrations (1.5~4.5 mol/L), in which the microscopic morphology of the treated copper foam surface is analyzed by scanning electron microscopy. The capillary experiments are carried out to quantify the wicking characteristics of the treated copper foams and the results are compared with theoretical models. A the microscope is used to detect the flow stratification characteristics of the capillary rise process. The results show that the best wicking ability is obtained for the oxidation of copper foam using 3.5 mol/L of NaOH and NaClO_2_ solution. Gravity plays a major role in defining the permeability and effective pore radius, while the effect of evaporation can be ignored. The formation of a fluid stratified interface between the unsaturated and saturated zone results in capillary performance degradation. The current study is important for understanding the flow transport in porous materials.

## 1. Introduction

Due to high thermal conductance and long-distance heat transport with a corresponding small temperature difference, heat pipes have been widely used in the thermal management of high power density devices, such as microelectronic devices, spacecraft, nuclear reactors, LED modules, etc. [1,2,3] The heat pipe is typically built with a vacuum chamber, a wick and working fluid. Water is widely used as the working fluid for its high latent heat and other desirable properties such as being contamination free and of low cost. The wick is used to carry liquid from the condenser side to the evaporator side, which is the core operation of the heat pipes and also provides microstructures for evaporation enhancement. Typical wicking materials for heat pipes include axial grooves [4,5], sintered metal powders [6,7], metal meshes [8,9], metal foams [10] and composite structures [11]. An axial grooved wick has a quite good permeability, while a low capillary pressure may limit it to be operated in the gravity vector orientation. Due to the size of the pore being at nano/micro scale, the permeability of sintered metal powder is generally low. The metal mesh wick is normally thin (<1 mm), which produces a moderate capillary pressure with a corresponding low permeability. An advanced composite wick can have a good capillary pressure and an acceptable permeability at the same time. However, it often requires complex manufacturing processes and has a poor stability. Metal foam has the characteristics of high capillary pumping capacity, large surface-to-volume ratio, good heat conductivity and high permeability. These properties make them promising candidates for heat pipes as wicks.

Many researchers have studied the thermal performance of metal foam-based heat pipes. Dhanabal et al. [12] tested the thermal performance of a flat heat pipe with a copper foam wick. The results showed that inserting the wick decreases the thermal resistance by 1.7~3.2% for varying heat input. Bao et al. [13] investigated the heat transfer characteristics of a metal foam multichannel heat pipe. Compared with the traditional pulsating heat pipe, the proposed metal foam multichannel heat pipe showed better heat transfer performance and a higher upper heat transfer limit. Shen et al. [14] investigated the effects of different metal foam structures on the heat transfer performance of microchannel heat sinks. The results suggested that the porosities have a small effect on the thermal performance but have a larger effect on the pressure drop. Tang et al. [15] used a wick structure comprised of multi-scale copper mesh and sintered particles to achieve a certain anti-gravity capacity. Rachedi & Chikh [16] numerically discussed the thermal characteristic of electronic cooling with foam. It was found that inserting a porous substrate with high thermal conductivity can reduce the temperature by 50% in comparison with a case without a porous material. Zhang et al. [17] developed a highly efficient cooling solution to the recently emerging high performance plasmonic solar cell technology by integrating an advanced nano-coated heat-pipe plate. The experimental results showed that the heat transportation capability is up to ten times higher than those of the metal plate and the conventional wickless heat-pipe plates. Xin et al. [18] studied the flow and thermal characteristics in a mini-grooved flat heat pipe.

However, one challenge for using metal foam as a wick in heat pipes is to improve its hydrophilicity. Low wettability of the wicking material directly reduces the critical heat load in a heat pipe [19]. Over the past years, alkali solutions such as NaOH, KOH, H_2_O_2_ and NH_3_/H_2_O have been employed on metal foams oxidation to create nano/micro-scale structures using a chemical immersion method [20,21,22,23,24]. Faghri [25] stated that copper oxides can reduce the wetting characteristics of a surface. Popova et al. [26] used a thin layer of CuO to enhance the wetting of a sintered copper wick. Wong and Lin [27] implied that the formation of Cu_2_O on the copper surface is the main factor for the loss of hydrophilicity. However, Yin et al. [28] suggested that volatile organic contaminations is the main reason for the loss of hydrophilicity. Shum et al. [29] developed a chemical blackening process for creating a superhydrophilic surface on copper foams. The results found that the contact angle of the treated foam reduced from 91° to 0° and remained unchanged for more than 3 months. Also, by compressing the treated foams from 2 mm to 0.5 mm, there was a 183% increase in equilibrium height. This study expands and extends our earlier work. The mechanism of wettability is still not fully understood in metal foam and an economical method that achieves superhydrophilicity on metal foam surfaces is also desirable.

The capillary performance of a wick, which is characterized by permeability and effective pore radius, is the key factor affecting the heat transfer performance and capillary limit of the heat pipes. Permeability is defined as the resistance against the liquid passing through porous material. A high permeability will result in a lower liquid pressure drop. Effective pore radius is a parameter used to describe the available pressure rise for liquid pumping, which is defined as the actual pore radius of a porous material divided by the cosine of the liquid contact angle on that material. To characterize the permeability and effective pore radius of the metal foam wick, a series of methods have been developed in the past two decades. By measuring the pressure drop characteristic of the metal foams, the permeability coefficients can be obtained by Darcy’s law or Forchheimer equations from experimental data [30,31,32,33,34]. However, these studies were developed in a small pore density of 5~40 pores per inch (PPI), which means the number of pores in one linear inch. Only Du Plessis et al. [35] reported the permeability values for 45 PPI, 60 PPI and 100 PPI metal foams. Therefore, the permeability data are still insufficient in metal foam with high PPI. Effective pore radius tests mainly include bubble point test and risen meniscus test [25]. The rate-of-rise test of liquid in a wick by measuring its wetting height visually or the increasing mass with a balance can extract both permeability and effective pore radius [36]. The capillary rise-of-rise method has been widely studied by Deng et al. [37], Feng et al. [38], and Jafari et al. [39]. Lucas and Washburn considered only the capillary force at the meniscus and the friction force inside a porous material in the capillary rise process. Balancing the two forces yields the commonly known Lucas–Washburn equation [40]. Several researchers tried to improve the Lucas–Washburn equation and introduced additional effects. Holley and Faghri [41] developed a new equation which permits the extraction of permeability and effective pore radius by adding gravitational effects to the Lucas–Washburn equation. Fries et al. [42] developed a fully implicit solution for all the effects except for inertia. Our latest studies have adopted different coefficients in the viscous and inertial terms of the Lucas–Washburn equation to account for the nonuniform velocity distribution for capillary flow in channels of different geometries [43,44].

In this study, seven different NaOH and NaClO_2_ solution concentrations (from 1.5 to 4.5 mol/L) are studied to create nano/micro-scale surface morphology on the surfaces of open-cell copper foam with high-PPI and to analyze their liquid wettability. Characterizing the permeability and effective pore radius in these open-cell copper foams by the capillary experiments is also carried out by comparing the experimental data with the theoretical models. Meanwhile, a microscope is used to detect the internal flow characteristics of copper foam wicks to reveal the flow stratification.

## 2. Methodology and Experimental Setup

### 2.1. Fabrication of Copper Foams Sample

Copper metal foams used in this study were purchased from Changsha Lyrun Material Co., Ltd., Changsha, China, and the detailed information is given in Table 1. For these copper metal foams, the PPI varies from 35 to 130, the porosity ranges from 0.91 to 0.97, and they have a thickness range of 0.8~2 mm. For measuring the porosity of copper metal foams, it was firstly cut as a sample with length and width (20 × 20 mm) by a laser cutting machine with an accuracy of 0.03 mm. A high precision analytical balance (Motic ES-23BZ) with a precision of 0.1 mg was then used to measure the sample mass. Thus, the porosity is determined by a density method:(1)$$\varepsilon =1-\frac{m_{c}/\rho_{c}}{A\delta }$$
where *ρ*_c_ is the density of copper (8.9 g/cm^3^), *A* is the cross-section area of the copper foam (400 mm^2^), and *m_c_* is the mass of the copper foam sample. The measurement of the mass of each copper foam sample was repeated four times and each type of sample was cut into three pieces. The average porosity was obtained using Equation (1).

In order to achieve a high and stable wicking performance using copper foams, a chemical oxidation approach was used to create a superhydrophilic surface on the copper foams. The processing procedure of copper foam oxidation was divided into the following steps: (1)The copper foam was first cleaned in an ultrasonic bath with deionized water for two intervals of 5 min and replaced with fresh deionized water in between the intervals.(2)The copper foam was then immersed in the acetic acid (99.9%) bath for 1 h at 40 °C and atmospheric conditions. The acid treatment was performed in a rectangular glass container using a stirrer at 250 rpm.(3)After that, the copper foams were flushed with nitrogen flow for 5 min and then dried in the vacuum oven operating at 1 kPa. The heating process was composed of a ramp from room temperature (20 °C) to 70 °C in 30 min and followed by a 30 min plateau at 70 °C.(4)Subsequently, the copper foams were placed into a solution of NaOH and NaClO_2_ at 90 °C for varying treatment concentrations. Since NaOH reacts with NaClO_2_ in a 1:1 molar stoichiometric ratio, the concentration of NaOH is equal to that of NaClO_2_. The blackening treatment was performed in a rectangular glass container using a stirrer at 250 rpm for 1 h.(5)After the blackening, the copper foams were flushed with nitrogen flow for 15 min and then dried at 250 °C in the vacuum oven at 1 kPa for 2 h. The heating process was composed of a ramp from room temperature (20 °C) to 250 °C in 1.2 h and followed by a 2 h plateau at 250 °C. The blackening copper foams were then stored in a vacuum desiccator until the next use.

### 2.2. Setup for Rate-of-Rise Experiment

A schematic drawing of the experimental setup to measure the rate of rising of capillary flow in the copper foams is shown in Figure 1. The capillary flow of deionized water through copper foam was studied by capturing the mass loss in the reservoir using a high precision analytical balance. Copper foam samples were cut into 200 mm long by 20-mm-wide strips. The sample was suspended in a vertical position and fixed by the holder. The liquid reservoir was placed on the analytical balance. Before the experiments, all the test samples were dried by vacuum oven. To perform a measurement, the precision lifting table was used to descend the sample and brought it into contact with the liquid. The balance measured a change in weight as the liquid was drawn into the porous structure at a rate of ten data per second. The balance was connected to a computer and the weight was recorded continuously as a function of time. The capillary rise process was recorded about twenty minutes into each test as the equilibrium height is close to being achieved. The tests were conducted at the cleanroom with constant temperature and humidity conditions, i.e., a temperature of 25 ± 0.1 °C and humidity of 58 ± 1%. Deionized water produced by the Thermo Scientific Smart2Pure UV/UF ultrapure water system was used as the working medium for all rate-of rise experiments. 

To evaluate the effect of the evaporation of deionized water, evaporations in the liquid reservoir and from the copper foam surface were measured. Firstly, the liquid reservoir filled with deionized water was placed on the balance, and mass reduction was recorded by the balance as a function of time. After each test, the copper foam sample was placed on the balance and mass reduction was also measured by the balance. Secondly, the mass of liquid evaporated per area and time (kg·m^−2^·s^−1^) from the liquid reservoir and the copper foam surface was calculated separately with the assumption that the evaporation rate is constant throughout the surface of the liquid reservoir and copper foam sample. The round reservoir has a radius of 50 mm and a depth of 20 mm. Finally, evaporation mass *M*_e_ from both reservoir and copper foam should be subtracted from the weight measured *M* by the balance to obtain the mass uptake of the copper foam *M*_h_ = *M* − *M*_e_.

## 3. Theoretical Model

During the capillary rate-of-rise experiment, a rise of deionized water in the copper foams occurs as the copper foam encounters the deionized water. The governing equation for the capillary rate-of-rise can be derived using the momentum balance, which major includes capillary forces, gravity force and viscous force. The capillary pressure, created due to the meniscus appearing at the liquid–vapor interface, will pull the liquid upwards in the copper foams. It is known as the Laplace–Young equation.
(2)$$\Delta p_{cap}=\frac{2\sigma \cos\theta }{r}=\frac{2\sigma }{r_{eff}},$$
where *σ* is the surface tension, *θ* is the contact angle formed between solid and liquid, *r* is the pore radius and *r_eff_* is the effective pore radius.

For vertical copper foam, there is a hydrostatic pressure drop created by the liquid column
(3)$$\Delta p_{h}=\rho gh,$$
where *g* is the gravitational acceleration, *ρ* is the fluid density, and *h* is the capillary rise height.

To calculate the viscous force, some assumptions are made, such as: (1) one-dimensional and uniform saturation with liquid along the wetted length, (2) viscous pressure drop is described by Darcy’s Law, in which the inertial effect is negligible, (3) evaporation is uniformly distributed as per area and time *m*_e_ (kg·m^−2^·s^−1^).

The mass balance of a wicking process with evaporation is shown in Figure 2. The total mass inflow is made up of two components at position z: the mass flow necessary to supply the movement of the liquid front *M*_h_ and the total evaporation mass flow *M*_e_. 

Based on Darcy’s law, the viscous pressure loss to supply the movement of the liquid front *M*_h_ is calculated as
(4)$$\Delta p_{f}=\frac{\varepsilon }{K}\mu h\frac{dh}{dt}.$$

The evaporation mass is given by
(5)$$M_{e}=m_{e}2\left(W+\delta \right)h.$$

Therefore, the refill velocity to refill the evaporated liquid is given by
(6)$$\upsilon_{r}=\frac{M_{e}}{\rho W\delta \varepsilon }=\frac{2m_{e}h\left(W+\delta \right)}{\rho W\delta \varepsilon }.$$

The viscous pressure loss to refill the evaporated liquid *M_e_*, which is height dependent, is calculated as
(7)$$\Delta p_{e}=\frac{\varepsilon }{K}\mu h\int_{0}^{h}\frac{2m_{e}z\left(W+\delta \right)}{\rho W\delta \varepsilon }dh=\frac{\mu m_{e}\left(W+\delta \right)}{K\rho W\delta }h^{2},$$
where *δ* is the foam thickness, *W* is the foam width, *m_e_* is the evaporation rate, *K* is the permeability, *ε* is the porosity and d*h*/d*t* is the capillary rise velocity.

Due to momentum balance, the capillary pressure is balanced by the gravity and viscous force as presented
(8)$$\Delta p_{cap}=\Delta p_{h}+\Delta p_{f}+\Delta p_{e}$$
and
(9)$$\frac{2\sigma }{r_{eff}}=\rho gh+\frac{\varepsilon }{K}\mu h\frac{dh}{dt}+\frac{\mu m_{e}\left(W+\delta \right)}{K\rho W\delta }h^{2}.$$

It can be transformed as following:(10)$$\frac{dh}{dt}=A\frac{1}{h}-B-Ch,$$
where the coefficients *A*, *B*, *C* are defined as,
(11)$$A=\frac{2\sigma }{\mu \varepsilon }\frac{K}{r_{eff}},$$
(12)$$B=\frac{\rho gK}{\mu \varepsilon }$$
and
(13)$$C=\frac{m_{e}\left(W+\delta \right)}{\rho W\delta \varepsilon }.$$

Starting from Equation (10), we can derive an analytic expression for the time needed to reach a certain height of the liquid front *t*(*h*). Rewriting Equation (10) gives
(14)$$\int_{0}^{h}\frac{h}{A-Bh-Ch^{2}}dh=\int_{0}^{t}dt.$$

Setting the initial condition *t* (*h*→0) = 0, Equation (14) is simplified for the following four cases:(1)When the gravitational and evaporation effects are neglected in Equation (14), *B* = *C* = 0, it is the Lucas-Washburn equation as
(15)$$t=\frac{h^{2}}{2A}.$$

(2)When the evaporation effect is neglected in Equation (14), *C* = 0, it leads to the following equation


(16)
$$t=\frac{-1}{B^{2}}\left[Bh+A\ln\left(1-\frac{B}{A}h\right)\right].$$


(3)When the gravitational effect is neglected in Equation (14), *B* = 0, it can be rewritten as


(17)
$$t=-\frac{1}{2C}\ln\left(1-\frac{C}{A}h^{2}\right).$$


(4)As gravity and evaporation are considered, the solution have to satisfy that $\psi =-4AC-B^{2}<0$, the final solution is 


(18)
$$\begin{array}{l} t=\frac{1}{2C}\left[-\ln\left(\frac{-Ch^{2}-Bh+A}{A}\right)\right]-\frac{B}{2C\sqrt{-\psi }}\\ \begin{array}{cc} & \times \ln\left[\frac{\left(-2Ch-B-\sqrt{-\psi }\right)\left(-B+\sqrt{-\psi }\right)}{\left(-2Ch-B+\sqrt{-\psi }\right)\left(-B-\sqrt{-\psi }\right)}\right] \end{array} \end{array}.$$


Therefore, *t*(*h*) is expressed explicitly as a function of capillary rise height *h*, liquid properties, foam geometry and two unknowns: permeability *K* and effective pore radius *r*_eff_. As stated before, the wicking height is calculated based on the assumption that the liquid is uniform saturation inside the copper foams. It can be calculated as the following equation
(19)$$h=\frac{m}{\rho \delta W\varepsilon }.$$
where *m* is the mass.

After measuring liquid rate-of-rise versus time, permeability *K* and effective pore radius *r*_eff_ can be determined by fitting the experimental data to the above equations. The value of permeability *K* and effective pore radius *r*_eff_ is obtained to minimize the error between the experimental data and predicted data by equations based on the least squares method.

The experimental uncertainties in the indirect measurements were evaluated by the procedure described by Moffat [45]. It is defined as following
(20)$$W={\left[\sum_{i=1}^{i=n}{\left(\frac{\partial f}{\partial x_{i}}\delta x_{i}\right)}^{2}\right]}^{0.5}.$$

In this study, experimental uncertainties are associated with measurements of length and mass. The accuracy of the laser cutting machine is 0.03 mm and the precision of the high precision analytical balance is 0.1 mg. The uncertainty in the porosity determined by Equation (1) has a maximum of 3.7%. Uncertainty in the capillary rise height calculated by Equation (19) has a maximum of 5.3%.

## 4. Results and Discussion

### 4.1. Surface Characteristics of Blackening Copper Foam

Figure 3 shows the SEM images at 50 magnifications of microscopic morphology of blackening copper foam. The Copper foam samples have a PPI of 130, thickness of 2.0 mm and porosity of 96.5% and are oxidized treatment with seven different NaOH and NaClO_2_ solution concentrations from 1.5 mol/L to 4.5 mol/L.

It is found from a naked eye observation that the black oxide coated is formed all over the copper foam surface inside out. The density of the black oxide coated grows continuously until the concentration of NaOH and NaClO_2_ solution reaches 3.5 mol/L, beyond which the density of the black oxide coated is reduced slightly.

The SEM images at 5 k magnification (insets at 30 k magnification) are shown in Figure 4. With a closer look at the micrograph images, it is seen that when the concentration of NaOH and NaClO_2_ solution increases to 3.0 mol/L, the ball-shape agglomerates are formed on the copper foam surface. The size of the ball-shape agglomerate is about 3 to 6 μm and it reaches its maximum size when the NaOH and NaClO_2_ solution concentration is 3.5 mol/L. The comparison of the inserts shown in Figure 4a,c indicates that the black oxide coated is a micro-sized sheet structure, while the thickness of the microsheets is in the range of nanometers. Compared with Figure 4a, the oxide layer thickness is increased rapidly, as the ball-shape agglomerates form that increase the surface roughness, as shown in Figure 4c. 

The effect of NaOH and NaClO_2_ solution concentrations on the capillary performance of the copper foam samples is further studied in Figure 5. It is revealed that the wicking heights all rise very quickly at the early stage of the capillary rising process and then slow down with the increase of time. With the increase of NaOH and NaClO_2_ solution concentrations, the maximum wicking height increases at first and then decreases. The best wicking ability is obtained for oxidation of copper foam using 3.5 mol/L of NaOH and NaClO_2_ solution. Based on the SEM images, copper foam oxidized in low NaOH and NaClO_2_ solution concentrations (1.5, 2.0 and 2.5 mol/L) shows the micro-sized sheets within extremely thin oxide layer thickness. However, as the solution concentration increases to 3.0 mol/L above, the surface morphology of copper foam is changed to micro-scale ball-shape agglomerates augmenting the surface roughness significantly. For a hydrophilic surface, the surface wettability can be enhanced by increasing the surface roughness [46]. Moreover, the gaps between ball-shape agglomerates can produce sufficient capillary suction to uptake the liquid. Therefore, it can be inferred that the formation of ball-shape agglomerates leads to a significant increase in the capillary performance of copper foam. In the following experiments, the copper foam samples will be oxidized in a NaOH and NaClO_2_ solution concentration of 3.5 mol/L. 

### 4.2. Analysis of Experiment Results

Figure 6 shows the raw wicking height, evaporation height and correction wicking height versus time for copper foam with PPI of 130, thickness of 2 mm and porosity of 96.5%. The raw wicking height rises rapidly at first and then tends to become smaller as the time increases. The reason for this may be due to the gravitational effect and the hygroscopicity of the copper foam. In the early stages, the capillary force in the copper foam sample is much larger than the flow resistance and gravity. Hence, the uptake height shows a fast rising velocity. However, as the amount of wicking mass increases, the gravity resistance and viscous force increase gradually, which results in an equilibrium state in capillary pressure, gravity and viscous force. At the initial stage, a jump is formed in wicking height. Based on the previous research by Shirazy et al. [47], the jump in mass includes wetting mass and wicking mass. The wetting mass is due to the formation of macroscopic meniscus on the contact between copper foam and liquid. The meniscus applies a pulling force to the samples due to the liquid surface tension, which results in a significant reduction in mass measured by the balance. Therefore, to obtain the wicking mass, the wetting mass should be subtracted from the raw mass data when the initial jump occurs. Meanwhile, the wetting mass can be recorded by detaching the copper foam from the liquid surface after each test. The mass of liquid evaporated per area and time (kg·m^−2^·s^−1^) from the liquid reservoir and the copper foam surface are calculated as the description in Section 2.2. In this case, the evaporation rate from the container and copper foam is obtained as 2.35 × 10^−5^ kg·m^−2^·s^−1^ and 8.63 × 10^−5^ kg·m^−2^·s^−1^, respectively. Therefore, the correction wicking mass is obtained by subtracting evaporation mass from the liquid reservoir and the copper foam surface and the wetting mass as the initial jump occurs from the raw measured mass in this study. According to the research by Shirazy et al. [47], the jump occurred in the order of 0.01 s. Therefore, the startup of the experiment was recognized when the jump occurred. 

In order to further study the meniscus effects, the raw wicking height and correction wicking height at the initial stage 0~10 s are shown in Figure 7. It can be seen that a jump is also observed in the correction wicking height. On the basis of visual observation, it is inferred that when the copper foam touches the test liquid, the meniscus will be formed around the end of the copper foam and its side surface due to the surface tension. However, after the copper foam detaches from the liquid surface, only the meniscus around the end of the copper foam disappears. Therefore, the side surface meniscus of the copper foam still remains, which causes the jump in the correction wicking height. Moreover, the side surface of the copper foam is large compared to its end (20 mm width compared to 0.2 mm thickness). In this case, a pulling force caused by macroscopic meniscus around the end of the copper foam and its side surface is 0.212 g and 0.604 g, which cause a height lift with 5.31 mm and 15.13 mm, respectively.

Based on the above-represented approaches, the correction wicking heights versus time for four copper foams with thickness of 2 mm and different PPI are shown in Figure 8. The liquid ascends rapidly through the copper foam during the first 50 s when the gravity and viscous force restraining liquid rise is much smaller than the capillary force lifting it up. The rise velocity then decreases and the lifting height gradually reaches stability. The wicking height increases as the increase of PPI and the best wicking height is obtained for copper foam with a PPI of 130. With the increase of PPI, a smaller void will be obtained which results in a larger capillary pressure and an associated greater flow resistance. In this case, it can be found that the effect of PPI on the capillary pressure is more significant than that of flow resistance.

To extract the permeability and effective pore radius of the four copper foams, Equation (14) with four cases will be analyzed based on the experimental data in Figure 8. However, considering that Equation (14) is established based on the law of momentum conservation, only the experimental data in the range of 1000 to 1200 s are used. Table 2 shows the calculated permeability, effective pore radius or the ratio of permeability and effective pore radius for copper foam with a PPI of 130. Using Equations (15) and (17), in which the effect of gravity and evaporation are neglected or that of gravity is neglected, the ratio of permeability and effective pore radius *K*/*r*_eff_ is obtained as 0.3342 μm and 0.3343 μm, respectively, with the root of mean square (RMS) value of 27.2% and 27.3%. When the effect of evaporation is neglected, the values of the permeability and effective pore radius are obtained as 982.5 μm^2^ and 250.48 μm, with an RMS value of 20.5%. As gravity and evaporation are considered, Equation (18) is too complicated to obtain the permeability and effective pore radius. Alternatively, the permeability and effective pore radius obtained by Equation (16) are checked using Equation (18) to examine the effect of evaporation. Figure 9 shows the predicted time calculated by Equation (18) versus the experimental data. The predicted times agree well with the experimental time with an RMS value of 21.7%. Therefore, it can be inferred that the effect of evaporation on the capillary performance parameters of copper foams can be neglected. However, the values of *K*/*r*_eff_ obtained by Equations (15) and (17) are significantly smaller than that obtained by Equation (18). It can be inferred that gravity plays an important role in the capillary performance parameters of copper foams. It is understandable that the gravity becomes a dominating force in the later time stages of capillary rise. Using Equation (17), the permeability and effective pore radius of the four copper foams with different PPI are obtained and summarized in Table 3.

### 4.3. Effect of Copper Foam Thickness

Figure 10 shows the wicking height versus time for copper foams with a PPI of 130 and different thicknesses. These copper foams have the thickness of 0.8, 1.0, 1.5 and 2.0 mm and porosities ranging from 0.914 to 0.965. The plot shows an increase in the wicking height with the increase of time. A similar capillary rise behavior is found in the copper foams with thickness of 0.8 mm and 1.0 mm, and with thicknesses of 1.5 and 2.0 mm. However, for copper foams with thicknesses of 0.8 and 1.0 mm, the wicking height is significantly higher than that of copper foams with thicknesses of 1.5 and 2.0 mm. 

Figure 11 illustrates the images at 1000 s of capillary rise processes in these four copper foams with different thicknesses. The liquid front line of the copper foams with thicknesses of 0.8 and 1.0 mm is significantly higher than that of the copper foams with thicknesses of 1.5 and 2.0 mm. The brightness of the copper foam surface is darkened gradually as the height increases for copper foams with thicknesses of 0.8 and 1.0 mm. Due to the fact that water reflects light more strongly than air, the reason for the darkening of the light is that the water content decreases with the increase of height in the copper foam surface. However, a bright dividing line is found in the lower part of copper foam with a thickness of 1.5 and 2.0 mm. 

In order to reveal the reason for the existence of a bright dividing line, a microscope is used to detect the characteristics of the capillary rise process. Figure 12 shows the typical microscope images at the 1000 s of capillary rise processes in the copper foam with a thickness of 2.0 mm. As expected, copper foam is completely dry above the upper dividing line (Figure 12a). Dry and wetting zones are observed around the upper dividing line (Figure 12b). Between the upper and lower dividing lines, the surface of skeleton of the copper foam is wetted by the water and it is filled by air between adjacent skeletons (Figure 12c). Hence, the liquid contents are less than the copper foam porosity in this zone, which can be defined as an unsaturated zone or partially saturated zone. The surface of copper foam includes complete and partial filling with liquid around the lower dividing line (Figure 12d). It is completed filled with liquid below the lower dividing line (Figure 12e). Therefore, the bright dividing line is the boundary between the saturated and unsaturated zone. The reason for the formation of a fluid stratified interface may be that as the flow resistance of the liquid on the side surface of copper foam is smaller than its inside, a thinner copper foam is more likely to absorb the liquid with the help of the meniscus on the side surface. However, with the increase of the thickness of copper foams, the advantage of the side macroscopic meniscus will be reduced. On the other hand, the size difference between pore and skeleton is cross scale, which makes it difficult for capillary liquid to completely fill the pores. Therefore, the formation of the dividing line in the copper foams may be attributed to the effect of the side macroscopic meniscus and macroporous skeleton structure. 

## 5. Conclusions

In the present study, the capillary rate-of-rise experiments are conducted to characterize the capillary performance of oxidized copper foam based on the measured mass method. The effects of NaOH and NaClO_2_ solution concentrations, macroscopic meniscus and copper foam thickness on the capillary flow characteristics of the oxidized copper foams are discussed. Moreover, the governing equations of the capillary model are established to obtain the permeability and effective pore radius of copper foams. The main conclusions are as follows:As the NaOH and NaClO_2_ solution concentration increases to 3.0 mol/L, the micro-scale ball-shape agglomerates are formed on the copper foam surface, which can significantly enhance the wettability performance of copper foam. The best wicking ability is obtained for oxidation of the copper foam using 3.5 mol/L of NaOH and NaClO_2_ solution.As the copper foam touches the liquid, the macroscopic meniscus is formed around the end and side surface of the copper foams. The meniscus applies a pulling force to the copper foam and results in a jump in wicking height at the initial stage using the measured mass method.Gravity plays a major role in defining the permeability and effective pore radius, while the effect of evaporation can be ignored. Permeability and effective pore radius of the four copper foams with PPI from 35 to 130 and thickness of 2.0 mm are obtained by fitting the theoretical model to the experimental data with an RMS of about 20%.Under the assumption of uniform saturation inside the copper foams, the maximum wicking height is obtained as 110 mm for thicknesses of 0.8 and 1.0 mm, and 60 mm for thicknesses of 1.5 mm and 2.0 mm in copper foams with a PPI of 130. A bright interface between the unsaturated and saturated zone is formed as the thickness of copper foams increases to 1.5 mm, which is considered the reason for wicking performance degradation. However, the effect of unsaturation to determine the permeability and effective pore radius of the copper metal needs to be studied further.

## Figures and Tables

**Figure 1 micromachines-13-02052-f001:**
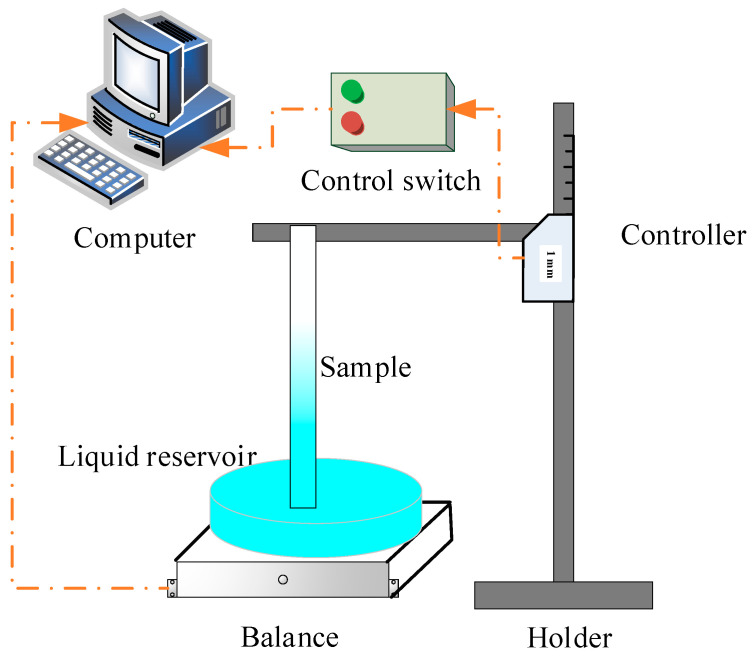
Schematic of the capillary rate-of-rise test apparatus.

**Figure 2 micromachines-13-02052-f002:**
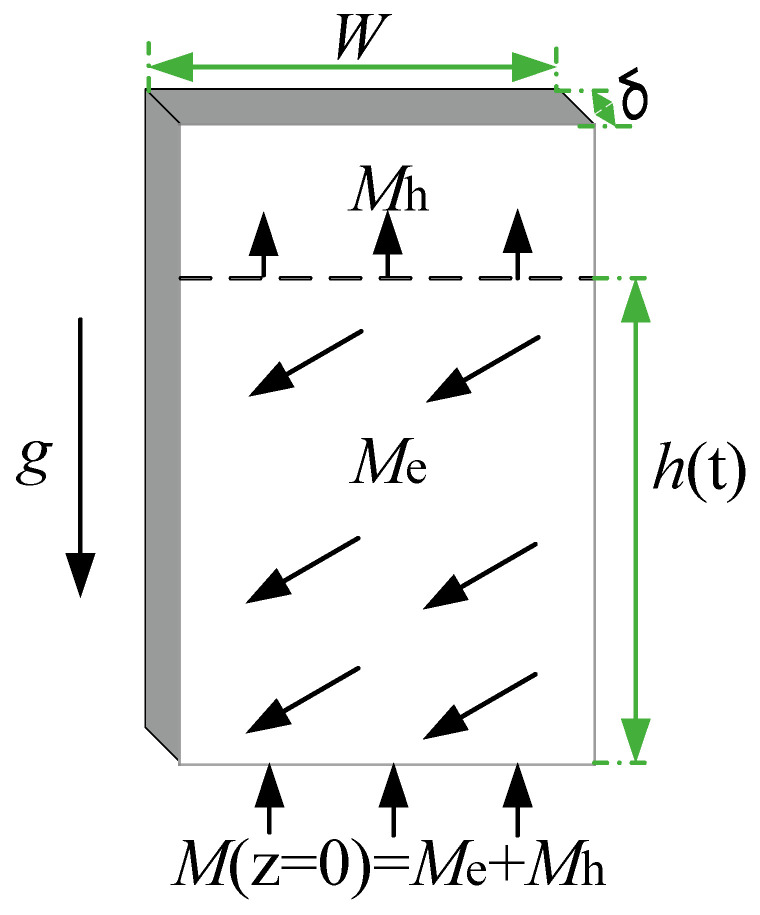
Mass balance of a wicking process with evaporation.

**Figure 3 micromachines-13-02052-f003:**
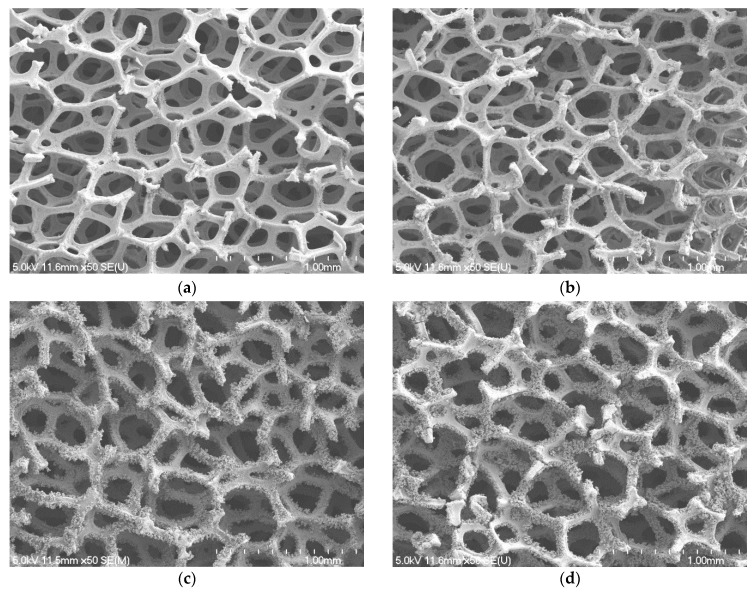
SEM images of blackening copper foam at 50 magnifications: (**a**) 1.5 mol/L; (**b**) 2.5 mol/L; (**c**) 3.5 mol/L; (**d**) 4.5 mol/L.

**Figure 4 micromachines-13-02052-f004:**
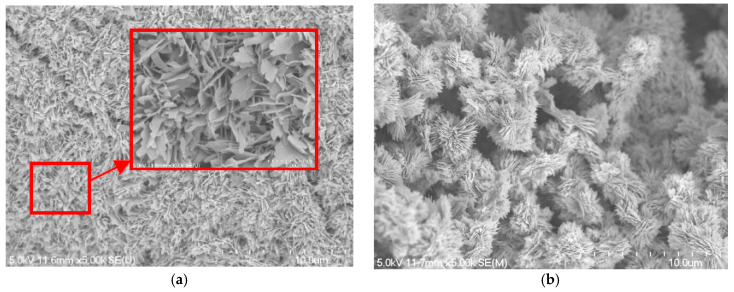
SEM images of blackening copper foam at 5 k magnifications: (**a**) 2.5 mol/L; (**b**) 3.0 mol/L; (**c**) 3.5 mol/L; (**d**) 4.5 mol/L.

**Figure 5 micromachines-13-02052-f005:**
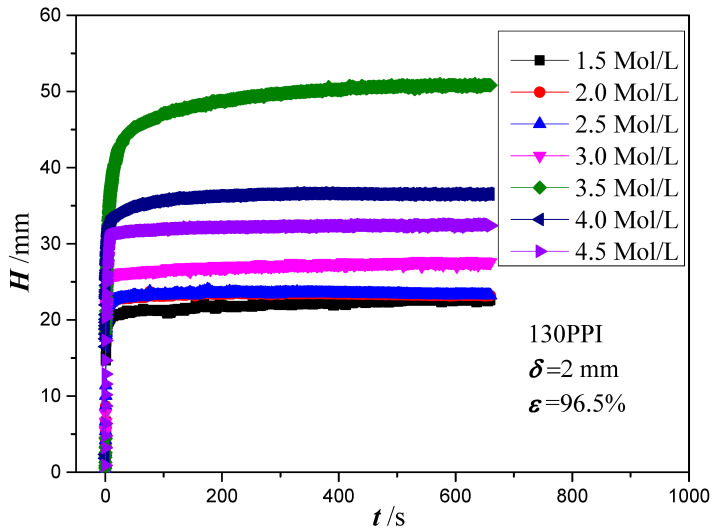
Wicking height versus time for different NaOH and NaClO_2_ solution concentrations.

**Figure 6 micromachines-13-02052-f006:**
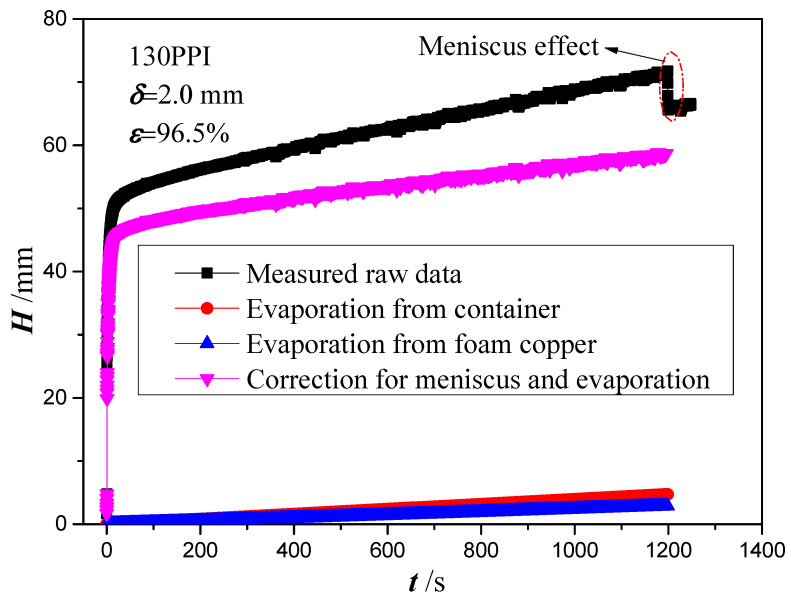
Raw data correction due to evaporation and meniscus effects.

**Figure 7 micromachines-13-02052-f007:**
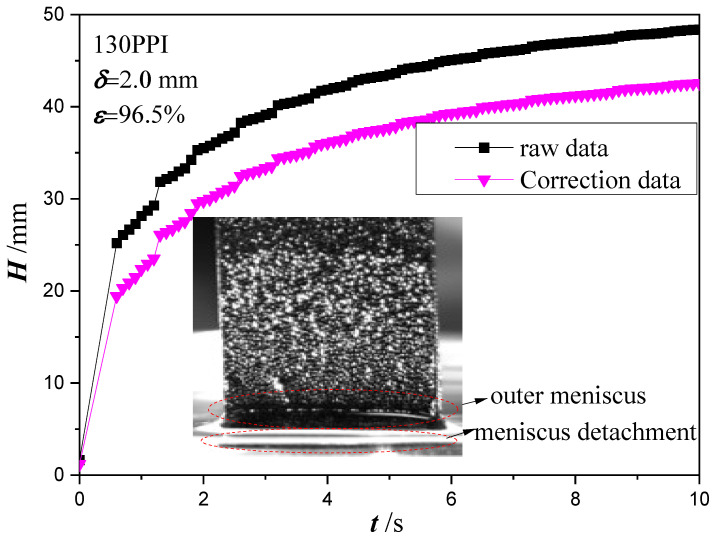
Wicking height at the initial stage 0~10 s.

**Figure 8 micromachines-13-02052-f008:**
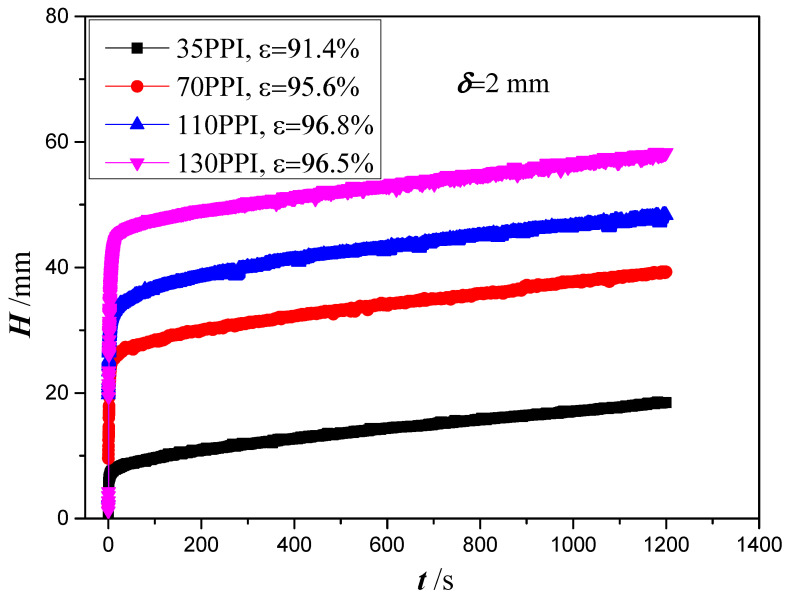
Wicking height versus time in copper foams with different PPI.

**Figure 9 micromachines-13-02052-f009:**
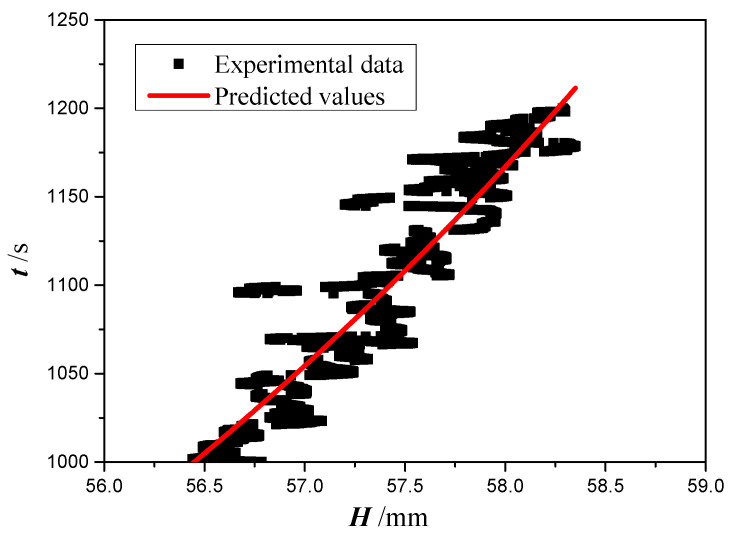
Comparison of time between experimental data and predicted values.

**Figure 10 micromachines-13-02052-f010:**
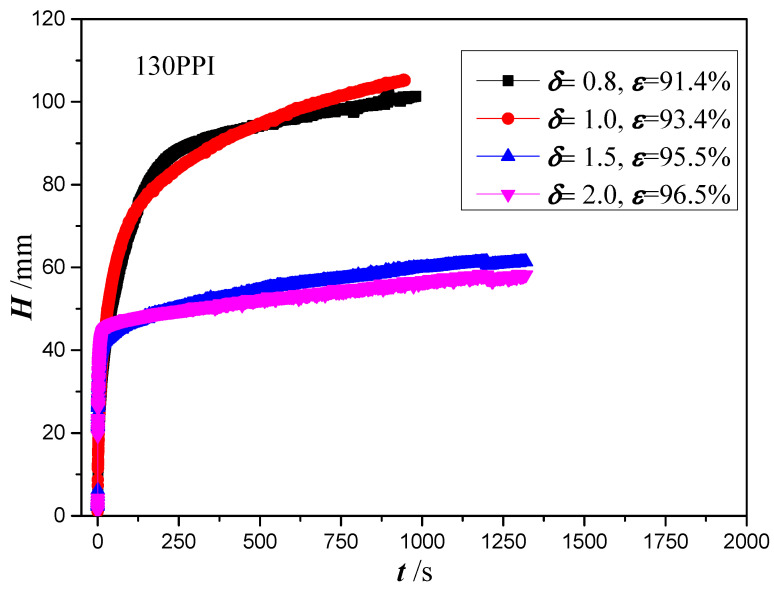
Wicking height versus copper foam with different thicknesses.

**Figure 11 micromachines-13-02052-f011:**
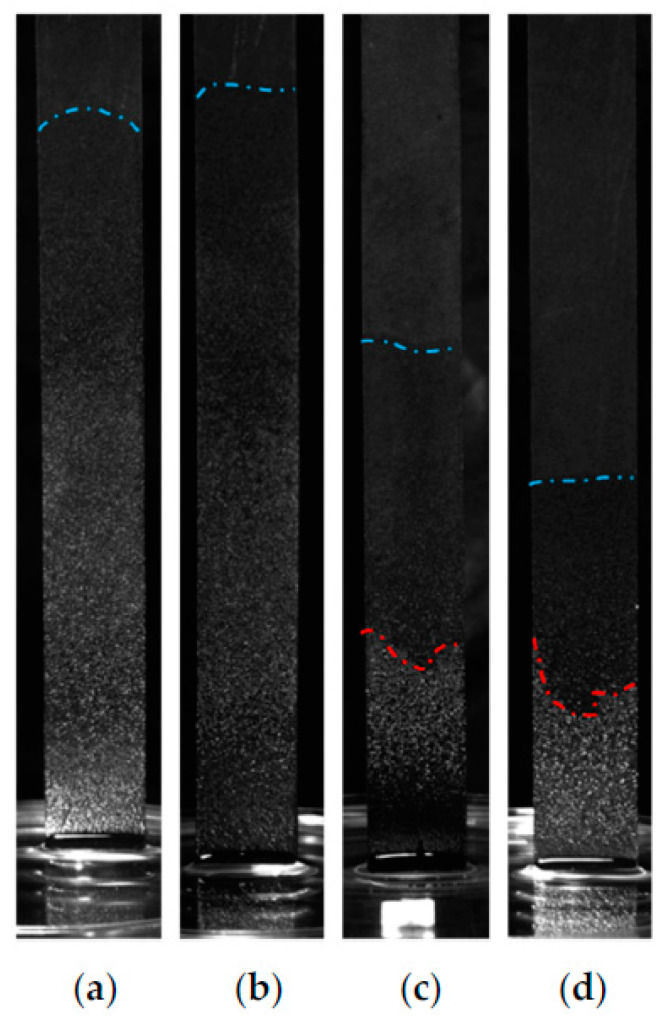
Images of capillary rise process of the four copper foams with different thicknesses at 1000 s: (**a**) 0.8 mm; (**b**) 1.0 mm; (**c**) 1.5 mm; (**d**) 2.0 mm, red line is the boundary between the saturated and unsaturated zone, and bule line the boundary between the unsaturated zone and the dry zone.

**Figure 12 micromachines-13-02052-f012:**
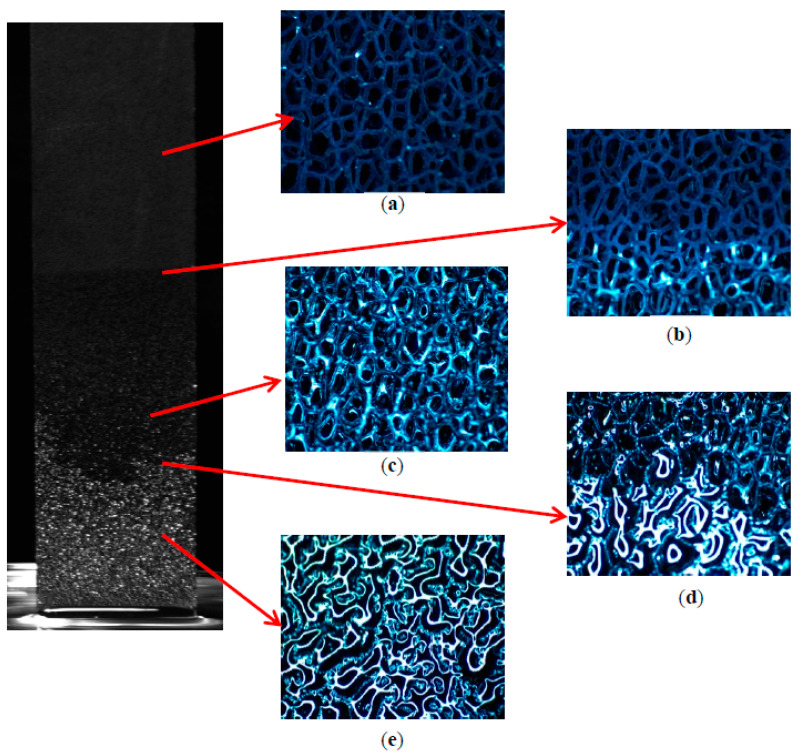
Microscope images of capillary rise process of the copper foam with a thickness of 2.0 mm at 1000 s: (**a**) drying zone; (**b**) upper dividing line; (**c**) unsaturated zone; (**d**) lower dividing line; (**e**) saturated zone.

**Table 1 micromachines-13-02052-t001:** Copper foam sample specifications.

PPI	Porosity ε	Thickness δ (mm)
35	91.4%	2
70	95.6%	2
110	96.8%	2
130	96.5%	2
130	95.5%	1.5
130	93.4%	1.0
130	91.4%	0.8

**Table 2 micromachines-13-02052-t002:** Capillary performance parameters in copper foam with PPI of 130.

Condition	*K* (μm^2^)	*r*_eff_ (μm)	*K*/*r*_eff_ (μm)	RMS
Ignore evaporation and gravity			0.3342	27.2%
Ignore evaporation	982.5	250.48	3.922	20.5%
Ignore gravity			0.3343	27.3%
No restriction	982.5	250.48	3.922	21.7%

**Table 3 micromachines-13-02052-t003:** Capillary performance parameters in copper foams with different PPI.

Sample	*ε*	*K* (μm^2^)	*r*_eff_ (μm)	*K*/*r*_eff_ (μm)	RMS
130 PPI	91.4%	982.5	250.48	3.922	20.5%
110 PPI	95.6%	1421.1	302.26	4.702	15.9%
70 PPI	96.8%	3403.5	371.64	9.158	18.6%
35 PPI	96.5%	7535.5	791.58	9.520	15.0%

## Data Availability

The datasets generated during and/or analyzed during the current study are available from the corresponding author on reasonable request.

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
