# Peer review of "Experimental Study on Capillary Microflows in High Porosity Open-Cell Metal Foams"

_micromachines, 2022, doi:10.3390/mi13122052_

Round 1
Reviewer 1 Report
In this paper, copper foams with superhydrophilic surface obtained by chemical blackening process are proposed. The influence of NaOH and NaClO2 solution concentrations, copper foam thickness, and pore size (PPI) on the capillary performance, permeability, and effective pore radius are studied. However, the questions below should be well addressed before publishing:
1. Some of the references are published in the earlier years, and more recent references are recommended.
2. In line 202-203, page 5, the definition of Mh and Me seems nor right, please check it again.
3. In the experimental setup, the mass difference of the copper foam is detected by the analytical balance. However, in the results and discussions section, all the capillary performance are characterized by height, why?
4. In line 344-345, page 11, it stated that a jump is formed at 0.5 s in the raw wicking height. But the reviewer think that this phenomenon will happen immediately when the copper foam contacted with the liquid, and the time of 0.5 s is just the test error.
5. In Fig.11, the reason why the upper dividing line and lower dividing line appeared is not stated.

Author Response
Response to Reviewers #1
We would like to thank the referees for their effort to review the paper and provide the valuable comments. The comments have been considered carefully and the manuscript has been revised carefully according to reviewers’ comments. Below are the replies addressing each specific comment:
Q1. Some of the references are published in the earlier years, and more recent references are recommended.
A: We thank the reviewer for the valuable comment. The references have been updated including some recent references in the revised manuscript.
Q2. In line 202-203, page 5, the definition of Mh and Me seems nor right, please check it again.
A: We thank the reviewer for the valuable comment. The definition of Mh and Me was retained. However, in line 184-186, the evaporation mass meva, the weight measured mtot and mass uptake mup were revised as Me, M and Mh, respectively, in the revised manuscript.
Q3. In the experimental setup, the mass difference of the copper foam is detected by the analytical balance. However, in the results and discussions section, all the capillary performance are characterized by height, why?
A: We thank the reviewer for the valuable comment. The governing equation for the capillary rate-of-rise includes capillary forces, gravity force and viscous force. To calculate the gravity force and viscous force, the capillary height needs to be known. Therefore, the measured mass needs to be converted into the wicking height to analyze gravity and viscous forces using Eq. (19). The description can refer to line 256-258 in the revised manuscript.
Q4. In line 344-345, page 11, it stated that a jump is formed at 0.5 s in the raw wicking height. But the reviewer think that this phenomenon will happen immediately when the copper foam contacted with the liquid, and the time of 0.5 s is just the test error.
A: We thank the reviewer for the valuable comment. The jump occurred in the order of 0.01 s in published literature (Mahmood R.S. Shirazy, International Journal of Heat and Mass Transfer, 2012). Therefore, in our experimental study, the time when the jump occurs is considered to be the startup of the experiment. The weight change at the initial stage 0 ~ 0.5 s is due to slight vibration of the platform caused by the sample drop-off. The description has been added by the highlight text in line 249-351 in the revised manuscript.
Q5: In Fig.11, the reason why the upper dividing line and lower dividing line appeared is not stated.
A: We thank the reviewer for the valuable comment. The formation of the dividing line in the copper foams may be attributed to the effect of the side macroscopic meniscus and macroporous skeleton structure. The reason has been explained by the highlight text in line 442-451 in the revised manuscript.

Reviewer 2 Report
The authors investigated the capillary flow in the metal foam with superhydrophilic surface treatment, which is very interesting. However, there are some issues needed to be handled before publication. The following are my comments:
1. The English language has many problems, which need to be improved.
2. What does PPI mean in describing metal foam?
3. As for the chemicals used, whether the concentration of NaOH is equal to that of NaClO_2? Please make it clear.
4. As for Figure 1, please draw the experimental setup completely including the lifting table and others if any.
5. As for the experimental setup, what are the dimensions of the water reservoir (e.g., width, length, and depth)? What is the initial submerged depth of the metal foam in water to test the capillary flow?
6. How to test the evaporation rate of the metal foam? Because the pores are open in the metal foam, it is difficult to contain water. Please make the test method clear.
7. In lines 175-178 on page 4, the authors assume the evaporation rate is constant. Please give the mass loss curve to illustrate the constant evaporation rate.
8. For equation 5, the meaning of W and δ should be given where they are first mentioned. In equation 7, there is no “z” within the integral symbol. In equation 19, the meaning of “m” should be given.
9. In equation 8, the pressure loss caused by evaporation may be caused by the curve water surface from the side face of metal foam. It may not be induced by the capillary force in the top waterfront, which needs to be reconsidered.
10. In Figure 5, why do high concentrations of the chemicals also lead to lower wicking height? Please give some explanation. In addition, how to measure the wicking heights should be given in the experimental section.
11. In Figure 5, the wicking height becomes stable after about 100s, but it is still not stable after 1000s in Figure 6. Please explain.
12. In lines 389-390 on page 13, the authors only mentioned the influence of gravity in the later stage of capillary rise, where the viscous force also plays an important role as the height increases. Lower wicking height at thicker metal foam shown in Figure 10 may be caused by the effect of viscous force.
13. In Figure 9, why there are so many heights at one time? Are these heights from repeated experiments? How many repeated experiments are carried out?
14. Figure 12 shows water is not saturated at high wicking height, which is not consistent with the assumptions (1-3) in lines 197-200 on page 5. How to handle this problem?
Author Response
Response to Reviewers #2
We would like to thank the referees for their effort to review the paper and provide the valuable comments. The comments have been considered carefully and the manuscript has been revised carefully according to reviewers’ comments. Below are the replies addressing each specific comment:
Q1. The English language has many problems, which need to be improved.
A: We appreciate the comments from the reviewer. We have carefully checked and revised the language in the revised manuscript.
Q2. What does PPI mean in describing metal foam?
A: We appreciate the comments from the reviewer. PPI (pores per inch) is the pore density in metal foam, which is defined as the number of pores in one linear inch. The description has been added by the highlight text in line 95 in the revised manuscript
Q3. As for the chemicals used, whether the concentration of NaOH is equal to that of NaClO_2? Please make it clear.
A: We appreciate the comments from the reviewer. Since NaOH reacts with NaClO2 in 1:1 molar stoichiometric ratio, the concentration of NaOH is equal to that of NaClO2. The description has been added by the highlight text in line 150-151 in the revised manuscript.
Q4. As for Figure 1, please draw the experimental setup completely including the lifting table and others if any.
A: We appreciate the comments from the reviewer. The experimental setup has revised in the revised manuscript.
Q5. As for the experimental setup, what are the dimensions of the water reservoir (e.g., width, length, and depth)? What is the initial submerged depth of the metal foam in water to test the capillary flow?
A: We appreciate the comments from the reviewer. The round reservoir has a radius of 50 mm and a depth of 20 mm. The description has been added by the highlight text in line 183-184 in the revised manuscript. The submerged depth of the metal foam is very small that it has not been measured. However, the effect of submerged depth can be eliminated. Since the wetting mass caused by macroscopic meniscus has been subtracted from the raw mass data.
Q6. How to test the evaporation rate of the metal foam? Because the pores are open in the metal foam, it is difficult to contain water. Please make the test method clear.
A: We appreciate the comments from the reviewer. Due to the liquid surface tension, the water can be easily absorbed on the surface of skeleton of the foam metal. Thus, leakage can’t occur. The calculation method about the evaporation rate from metal foam can be found in Section 2.2. The key to calculate the evaporation rate is to determine the mass of liquid evaporated per area and time (kg·m−2·s−1) from the copper foam surface.
Q7. In lines 175-178 on page 4, the authors assume the evaporation rate is constant. Please give the mass loss curve to illustrate the constant evaporation rate.
A: We appreciate the comments from the reviewer. The tests were conducted at the cleanroom with constant temperature and humidity conditions. Thus, the effect of environment can be ignored. It is reasonable to assume that the evaporation rate per area and time is constant. The evaporation mass from the reservoir versus time is shown as follows.
Q8. For equation 5, the meaning of W and δ should be given where they are first mentioned. In equation 7, there is no “z” within the integral symbol. In equation 19, the meaning of “m” should be given.
A: We appreciate the comments from the reviewer. W and δ has been added in Fig. 2. Z has been changed as h in equation 7. And the description of m has been added in the revised manuscript.
Q9. In equation 8, the pressure loss caused by evaporation may be caused by the curve water surface from the side face of metal foam. It may not be induced by the capillary force in the top waterfront, which needs to be reconsidered.
A: We appreciate the comments from the reviewer. In equation 5, we can find that the evaporation mass is formed around the metal foam, which is calculated as
. (5)
So, the definition of the viscous pressure loss caused by evaporation is right in the study. In fact, this formula has been adopted by some published literatures to analyze the effect of evaporation.
Q10. In Figure 5, why do high concentrations of the chemicals also lead to lower wicking height? Please give some explanation. In addition, how to measure the wicking heights should be given in the experimental section.
A: We appreciate the comments from the reviewer. It can be found from a naked eye observation that the density of the black oxide coated grows continuously until the concentration of NaOH and NaClO2 solution reaches 3.5 Mol/L, beyond which the density of the black oxide coated is reduced slightly. Therefore, the best wicking ability can be obtained for oxidation of copper foam using 3.5 Mol/L of NaOH and NaClO2 solution. The balance was interfaced to a computer and the weight was recorded continuously as a function of time. Thus, the wicking height as a function of time can be obtained using equation 19.
Q11. In Figure 5, the wicking height becomes stable after about 100s, but it is still not stable after 1000s in Figure 6. Please explain.
A: At the late stage, an equilibrium state in capillary pressure, gravity and viscous force was formed. According to the previous research, it will take several hours to reach stability. Therefore, it can be more believable that the wicking height will continue to increase very slowly with an equilibrium state.
Q12. In lines 389-390 on page 13, the authors only mentioned the influence of gravity in the later stage of capillary rise, where the viscous force also plays an important role as the height increases. Lower wicking height at thicker metal foam shown in Figure 10 may be caused by the effect of viscous force.
A: We appreciate the comments from the reviewer. Considering the effect of evaporation and gravity, we obtained that gravity plays an important role in the capillary performance parameters of copper foams, while the effect of evaporation can be neglected. In fact, viscous force has always been the most important factor to define the permeability and effective pore radius of copper foams. The reason for lower wicking height at thicker metal foam has been explained by the highlight text in line 442-451 in the revised manuscript.
Q13. In Figure 9, why there are so many heights at one time? Are these heights from repeated experiments? How many repeated experiments are carried out?
A: We appreciate the comments from the reviewer. In Fig. 19, one time corresponds to one height. However, we obtained 10 data points per second. As a result, there are very large data points and some data points can fluctuate in adjacent time periods. However, the change trend of the experiment points can be used to obtain the permeability and effective pore radius of copper foams.
Q14. Figure 12 shows water is not saturated at high wicking height, which is not consistent with the assumptions (1-3) in lines 197-200 on page 5. How to handle this problem?
A: We appreciate the comments from the reviewer. To obtain the permeability and effective pore radius of copper foams, the wicking height was obtained with the assumption that the liquid is uniform saturation inside the copper foams. However, the effect of unsaturation to determine the permeability and effective pore radius has not been reported in the published literature. We need to further study this problem in our future studies. The description has been added in conclusions in the revised manuscript.
